# ALICE: Towards Understanding Adversarial Learning for Joint Distribution Matching

**Chunyuan Li[1], Hao Liu[2], Changyou Chen[3], Yunchen Pu[1], Liqun Chen[1], Ricardo Henao[1] and Lawrence Carin[1]**
[1]Duke University   [2]Nanjing University   [3]University at Buffalo
cl319@duke.edu

## Abstract

We investigate the non-identifiability issues associated with bidirectional adversarial training for joint distribution matching. Within a framework of conditional entropy, we propose both adversarial and non-adversarial approaches to learn desirable matched joint distributions for unsupervised and supervised tasks. We unify a broad family of adversarial models as joint distribution matching problems. Our approach stabilizes learning of unsupervised bidirectional adversarial learning methods. Further, we introduce an extension for semi-supervised learning tasks. Theoretical results are validated in synthetic data and real-world applications.

## 1   Introduction

Deep directed generative models are a powerful framework for modeling complex data distributions. Generative Adversarial Networks (GANs) [1] can implicitly learn the data generating distribution; more specifically, GAN can learn to sample from it. In order to do this, GAN trains a *generator* to mimic real samples, by learning a mapping from a latent space (where the samples are easily drawn) to the data space. Concurrently, a *discriminator* is trained to distinguish between generated and real samples. The key idea behind GAN is that if the discriminator finds it difficult to distinguish real from artificial samples, then the generator is likely to be a good approximation to the true data distribution.

In its standard form, GAN only yields a *one-way* mapping, *i.e.,* it lacks an inverse mapping mechanism (from data to latent space), preventing GAN from being able to do inference. The ability to compute a posterior distribution of the latent variable conditioned on a given observation may be important for data interpretation and for downstream applications (*e.g.,* classification from the latent variable) [2, 3, 4, 5, 6, 7]. Efforts have been made to simultaneously learn an efficient bidirectional model that can produce high-quality samples for both the latent and data spaces [3, 4, 8, 9, 10, 11]. Among them, the recently proposed Adversarially Learned Inference (ALI) [4, 10] casts the learning of such a bidirectional model in a GAN-like adversarial framework. Specifically, a discriminator is trained to distinguish between two joint distributions: that of the real data sample and its inferred latent code, and that of the real latent code and its generated data sample.

While ALI is an inspiring and elegant approach, it tends to produce reconstructions that are not necessarily faithful reproductions of the inputs [4]. This is because ALI only seeks to match two joint distributions, but the dependency structure (correlation) between the two random variables (conditionals) within each joint is *not* specified or constrained. In practice, this results in solutions that satisfy ALI's objective and that are able to produce real-looking samples, but have difficulties reconstructing observed data [4]. ALI also has difficulty discovering the correct pairing relationship in domain transformation tasks [12, 13, 14].

In this paper, (*i*) we first describe the *non-identifiability* issue of ALI. To solve this problem, we propose to regularize ALI using the framework of *Conditional Entropy* (CE), hence we call the proposed approach ALICE. (*ii*) Adversarial learning schemes are proposed to estimate the conditional

entropy, for both unsupervised and supervised learning paradigms. (*iii*) We provide a unified view for a family of recently proposed GAN models from the perspective of joint distribution matching, including ALI [4, 10], CycleGAN [12, 13, 14] and Conditional GAN [15]. (*iv*) Extensive experiments on synthetic and real data demonstrate that ALICE is significantly more stable to train than ALI, in that it consistently yields more viable solutions (good generation and good reconstruction), without being too sensitive to perturbations of the model architecture, *i.e.,* hyperparameters. We also show that ALICE results in more faithful image reconstructions. (*v*) Further, our framework can leverage paired data (when available) for semi-supervised tasks. This is empirically demonstrated on the discovery of relationships for cross domain tasks based on image data.

## 2    Background

Consider two general marginal distributions $q(\boldsymbol{x})$ and $p(\boldsymbol{z})$ over $\boldsymbol{x} \in \mathcal{X}$ and $\boldsymbol{z} \in \mathcal{Z}$. One domain can be inferred based on the other using conditional distributions, $q(\boldsymbol{z}|\boldsymbol{x})$ and $p(\boldsymbol{x}|\boldsymbol{z})$. Further, the combined structure of both domains is characterized by joint distributions $q(\boldsymbol{x}, \boldsymbol{z}) = q(\boldsymbol{x})q(\boldsymbol{z}|\boldsymbol{x})$ and $p(\boldsymbol{x}, \boldsymbol{z}) = p(\boldsymbol{z})p(\boldsymbol{x}|\boldsymbol{z})$.

To generate samples from these random variables, adversarial methods [1] provide a sampling mechanism that only requires gradient backpropagation, without the need to specify the conditional densities. Specifically, instead of sampling directly from the desired conditional distribution, the random variable is generated as a deterministic transformation of two inputs, the variable in the source domain, and an independent noise, *e.g.,* a Gaussian distribution. Without loss of generality, we use an universal distribution approximator specification [9], *i.e.,* the sampling procedure for conditionals $\tilde{\boldsymbol{x}} \sim p_{\boldsymbol{\theta}}(\boldsymbol{x}|\boldsymbol{z})$ and $\tilde{\boldsymbol{z}} \sim q_{\boldsymbol{\phi}}(\boldsymbol{z}|\boldsymbol{x})$ is carried out through the following two generating processes:

$$\tilde{\boldsymbol{x}} = g_{\boldsymbol{\theta}}(\boldsymbol{z}, \boldsymbol{\epsilon}), \ \boldsymbol{z} \sim p(\boldsymbol{z}), \ \boldsymbol{\epsilon} \sim \mathcal{N}(0, \mathbf{I}), \text{ and } \tilde{\boldsymbol{z}} = g_{\boldsymbol{\phi}}(\boldsymbol{x}, \boldsymbol{\zeta}), \ \boldsymbol{x} \sim q(\boldsymbol{x}), \ \boldsymbol{\zeta} \sim \mathcal{N}(0, \mathbf{I}), \quad (1)$$

where $g_{\boldsymbol{\theta}}(\cdot)$ and $g_{\boldsymbol{\phi}}(\cdot)$ are two generators, specified as neural networks with parameters $\boldsymbol{\theta}$ and $\boldsymbol{\phi}$, respectively. In practice, the inputs of $g_{\boldsymbol{\theta}}(\cdot)$ and $g_{\boldsymbol{\phi}}(\cdot)$ are simple concatenations, $[\boldsymbol{z} \ \boldsymbol{\epsilon}]$ and $[\boldsymbol{x} \ \boldsymbol{\zeta}]$, respectively. Note that (1) implies that $p_{\boldsymbol{\theta}}(\boldsymbol{x}|\boldsymbol{z})$ and $q_{\boldsymbol{\phi}}(\boldsymbol{z}|\boldsymbol{x})$ are parameterized by $\boldsymbol{\theta}$ and $\boldsymbol{\phi}$ respectively, hence the subscripts.

The goal of GAN [1] is to match the marginal $p_{\boldsymbol{\theta}}(\boldsymbol{x}) = \int p_{\boldsymbol{\theta}}(\boldsymbol{x}|\boldsymbol{z})p(\boldsymbol{z})\mathrm{d}\boldsymbol{z}$ to $q(\boldsymbol{x})$. Note that $q(\boldsymbol{x})$ denotes the true distribution of the data (from which we have samples) and $p(\boldsymbol{z})$ is specified as a simple parametric distribution, *e.g.,* isotropic Gaussian. In order to do the matching, GAN trains a $\boldsymbol{\omega}$-parameterized adversarial discriminator network, $f_{\boldsymbol{\omega}}(\boldsymbol{x})$, to distinguish between samples from $p_{\boldsymbol{\theta}}(\boldsymbol{x})$ and $q(\boldsymbol{x})$. Formally, the minimax objective of GAN is given by the following expression:

$$\min_{\boldsymbol{\theta}} \max_{\boldsymbol{\omega}} \ \mathcal{L}_{\mathrm{GAN}}(\boldsymbol{\theta}, \boldsymbol{\omega}) = \mathbb{E}_{\boldsymbol{x} \sim q(\boldsymbol{x})}[\log \sigma(f_{\boldsymbol{\omega}}(\boldsymbol{x}))] + \mathbb{E}_{\tilde{\boldsymbol{x}} \sim p_{\boldsymbol{\theta}}(\boldsymbol{x}|\boldsymbol{z}), \boldsymbol{z} \sim p(\boldsymbol{z})}[\log(1 - \sigma(f_{\boldsymbol{\omega}}(\tilde{\boldsymbol{x}})))], \quad (2)$$

where $\sigma(\cdot)$ is the sigmoid function. The following lemma characterizes the solutions of (2) in terms of marginals $p_{\boldsymbol{\theta}}(\boldsymbol{x})$ and $q(\boldsymbol{x})$.

**Lemma 1 ([1])** *The optimal decoder and discriminator, parameterized by $\{\boldsymbol{\theta}^*, \boldsymbol{\omega}^*\}$, correspond to a saddle point of the objective in* (2)*, if and only if $p_{\boldsymbol{\theta}^*}(\boldsymbol{x}) = q(\boldsymbol{x})$.*

Alternatively, ALI [4] matches the *joint* distributions $p_{\boldsymbol{\theta}}(\boldsymbol{x}, \boldsymbol{z}) = p_{\boldsymbol{\theta}}(\boldsymbol{x}|\boldsymbol{z})p(\boldsymbol{z})$ and $q_{\boldsymbol{\phi}}(\boldsymbol{x}, \boldsymbol{z}) = q(\boldsymbol{x})q_{\boldsymbol{\phi}}(\boldsymbol{z}|\boldsymbol{x})$, using an adversarial discriminator network similar to (2), $f_{\boldsymbol{\omega}}(\boldsymbol{x}, \boldsymbol{z})$, parameterized by $\boldsymbol{\omega}$. The minimax objective of ALI can be then written as

$$\min_{\boldsymbol{\theta}, \boldsymbol{\phi}} \max_{\boldsymbol{\omega}} \ \mathcal{L}_{\mathrm{ALI}}(\boldsymbol{\theta}, \boldsymbol{\phi}, \boldsymbol{\omega}) = \mathbb{E}_{\boldsymbol{x} \sim q(\boldsymbol{x}), \tilde{\boldsymbol{z}} \sim q_{\boldsymbol{\phi}}(\boldsymbol{z}|\boldsymbol{x})}[\log \sigma(f_{\boldsymbol{\omega}}(\boldsymbol{x}, \tilde{\boldsymbol{z}}))]$$
$$+ \mathbb{E}_{\tilde{\boldsymbol{x}} \sim p_{\boldsymbol{\theta}}(\boldsymbol{x}|\boldsymbol{z}), \boldsymbol{z} \sim p(\boldsymbol{z})}[\log(1 - \sigma(f_{\boldsymbol{\omega}}(\tilde{\boldsymbol{x}}, \boldsymbol{z})))]. \quad (3)$$

**Lemma 2 ([4])** *The optimum of the two generators and the discriminator with parameters $\{\boldsymbol{\theta}^*, \boldsymbol{\phi}^*, \boldsymbol{\omega}^*\}$ form a saddle point of the objective in* (3)*, if and only if $p_{\boldsymbol{\theta}^*}(\boldsymbol{x}, \boldsymbol{z}) = q_{\boldsymbol{\phi}^*}(\boldsymbol{x}, \boldsymbol{z})$.*

From Lemma 2, if a solution of (3) is achieved, it is guaranteed that all marginals and conditional distributions of the pair $\{\boldsymbol{x}, \boldsymbol{z}\}$ match. Note that this implies that $q_{\boldsymbol{\phi}}(\boldsymbol{z}|\boldsymbol{x})$ and $p_{\boldsymbol{\theta}}(\boldsymbol{z}|\boldsymbol{x})$ match; however, (3) imposes *no restrictions* on these two conditionals. This is key for the identifiability issues of ALI described below.

## 3    Adversarial Learning with Information Measures

The relationship (mapping) between random variables $\boldsymbol{x}$ and $\boldsymbol{z}$ is not specified or constrained by ALI. As a result, it is possible that the matched distribution $\pi(\boldsymbol{x}, \boldsymbol{z}) \triangleq p_{\boldsymbol{\theta}^*}(\boldsymbol{x}, \boldsymbol{z}) = q_{\boldsymbol{\phi}^*}(\boldsymbol{x}, \boldsymbol{z})$ is undesirable for a given application.

To illustrate this issue, Figure 1 shows all solutions (saddle points) to the ALI objective on a simple toy problem. The data and latent random variables can take two possible values, $\mathcal{X} = \{x_1, x_2\}$ and $\mathcal{Z} = \{z_1, z_2\}$, respectively. In this case, their marginals $q(\boldsymbol{x})$ and $p(\boldsymbol{z})$ are known, *i.e.*, $q(\boldsymbol{x} = x_1) = 0.5$ and $p(\boldsymbol{z} = z_1) = 0.5$. The matched joint distribution, $\pi(\boldsymbol{x}, \boldsymbol{z})$, can be represented as a $2 \times 2$ contingency table. Figure 1(a) represents all possible solutions of the ALI objective in (3), for any $\delta \in [0, 1]$. Figures 1(b) and 1(c) represent opposite extreme solutions when $\delta = 1$ and $\delta = 0$, respectively. Note that although we can generate "realistic" values of $\boldsymbol{x}$ from any sample of $p(\boldsymbol{z})$, for $0 < \delta < 1$, we will have poor reconstruction ability since the sequence $\boldsymbol{x} \sim q(\boldsymbol{x})$, $\tilde{\boldsymbol{z}} \sim q_\phi(\boldsymbol{z}|\boldsymbol{x})$, $\tilde{\boldsymbol{x}} \sim p_\theta(\boldsymbol{x}|\tilde{\boldsymbol{z}})$, can easily

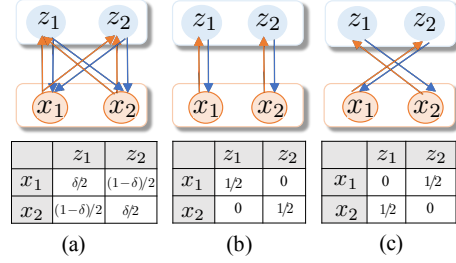

|       | $z_1$        | $z_2$        |
|-------|--------------|--------------|
| $x_1$ | $\delta/2$   | $(1-\delta)/2$ |
| $x_2$ | $(1-\delta)/2$ | $\delta/2$   |

(a)

|       | $z_1$ | $z_2$ |
|-------|-------|-------|
| $x_1$ | 1/2   | 0     |
| $x_2$ | 0     | 1/2   |

(b)

|       | $z_1$ | $z_2$ |
|-------|-------|-------|
| $x_1$ | 0     | 1/2   |
| $x_2$ | 1/2   | 0     |

(c)

Figure 1: Illustration of possible solutions to the ALI objective. The first row shows the mappings between two domains, The second row shows matched joint distribution, $\pi(\boldsymbol{x}, \boldsymbol{z})$, as contingency tables parameterized by $\delta = [0, 1]$.

result in $\tilde{\boldsymbol{x}} \neq \boldsymbol{x}$. The two (trivial) exceptions where the model can achieve perfect reconstruction correspond to $\delta = \{1, 0\}$, and are illustrated in Figures 1(b) and 1(c), respectively. From this simple example, we see that due to the flexibility of the joint distribution, $\pi(\boldsymbol{x}, \boldsymbol{z})$, it is quite likely to obtain an undesirable solution to the ALI objective. For instance, $i$) one with poor reconstruction ability or $ii$) one where a single instance of $\boldsymbol{z}$ can potentially map to any possible value in $\mathcal{X}$, *e.g.*, in Figure 1(a) with $\delta = 0.5$, $z_1$ can generate either $x_1$ or $x_2$ with equal probability.

Many applications require meaningful mappings. Consider two scenarios:

- **A1:** In unsupervised learning, one desirable property is *cycle-consistency* [12], meaning that the inferred $\boldsymbol{z}$ of a corresponding $\boldsymbol{x}$, can reconstruct $\boldsymbol{x}$ itself with high probability. In Figure 1 this corresponds to either $\delta \to 1$ or $\delta \to 0$, as in Figures 1(b) and 1(c).

- **A2:** In supervised learning, the pre-specified correspondence between samples imposes restrictions on the mapping between $\boldsymbol{x}$ and $\boldsymbol{z}$, *e.g.*, in image tagging, $\boldsymbol{x}$ are images and $\boldsymbol{z}$ are tags. In this case, paired samples from the desired joint distribution are usually available, thus we can leverage this supervised information to resolve the ambiguity between Figure 1(b) and (c).

From our simple example in Figure 1, we see that in order to alleviate the identifiability issues associated with the solutions to the ALI objective, we have to impose constraints on the conditionals $q_\phi(\boldsymbol{z}|\boldsymbol{x})$ and $p_\theta(\boldsymbol{z}|\boldsymbol{x})$. Furthermore, to fully mitigate the identifiability issues we require supervision, *i.e.*, paired samples from domains $\mathcal{X}$ and $\tilde{\mathcal{Z}}$.

To deal with the problem of undesirable but matched joint distributions, below we propose to use an information-theoretic measure to regularize ALI. This is done by controlling the "uncertainty" between pairs of random variables, *i.e.*, $\boldsymbol{x}$ and $\boldsymbol{z}$, using conditional entropies.

## 3.1 Conditional Entropy

Conditional Entropy (CE) is an information-theoretic measure that quantifies the uncertainty of random variable $\boldsymbol{x}$ when conditioned on $\boldsymbol{z}$ (or the other way around), under joint distribution $\pi(\boldsymbol{x}, \boldsymbol{z})$:

$$H^\pi(\boldsymbol{x}|\boldsymbol{z}) \triangleq -\mathbb{E}_{\pi(\boldsymbol{x},\boldsymbol{z})}[\log \pi(\boldsymbol{x}|\boldsymbol{z})], \text{ and } H^\pi(\boldsymbol{z}|\boldsymbol{x}) \triangleq -\mathbb{E}_{\pi(\boldsymbol{x},\boldsymbol{z})}[\log \pi(\boldsymbol{z}|\boldsymbol{x})]. \quad (4)$$

The uncertainty of $\boldsymbol{x}$ given $\boldsymbol{z}$ is linked with $H^\pi(\boldsymbol{x}|\boldsymbol{z})$; in fact, $H^\pi(\boldsymbol{x}|\boldsymbol{z}) = 0$ if only if $\boldsymbol{x}$ is a deterministic mapping of $\boldsymbol{z}$. Intuitively, by controlling the uncertainty of $q_\phi(\boldsymbol{z}|\boldsymbol{x})$ and $p_\theta(\boldsymbol{z}|\boldsymbol{x})$, we can restrict the solutions of the ALI objective to joint distributions whose mappings result in better reconstruction ability. Therefore, we propose to use the CE in (4), denoted as $\mathcal{L}_{\mathrm{CE}}^\pi(\boldsymbol{\theta}, \boldsymbol{\phi}) = H^\pi(\boldsymbol{x}|\boldsymbol{z})$ or $H^\pi(\boldsymbol{z}|\boldsymbol{x})$ (depending on the task; see below), as a regularization term in our framework, termed *ALI with Conditional Entropy* (ALICE), and defined as the following minimax objective:

$$\min_{\boldsymbol{\theta}, \boldsymbol{\phi}} \max_{\boldsymbol{\omega}} \mathcal{L}_{\mathrm{ALICE}}(\boldsymbol{\theta}, \boldsymbol{\phi}, \boldsymbol{\omega}) = \mathcal{L}_{\mathrm{ALI}}(\boldsymbol{\theta}, \boldsymbol{\phi}, \boldsymbol{\omega}) + \mathcal{L}_{\mathrm{CE}}^\pi(\boldsymbol{\theta}, \boldsymbol{\phi}). \quad (5)$$

$\mathcal{L}_{\mathrm{CE}}^\pi(\boldsymbol{\theta}, \boldsymbol{\phi})$ is dependent on the underlying distributions for the random variables, parametrized by $(\boldsymbol{\theta}, \boldsymbol{\phi})$, as made clearer below. Ideally, we could select the desirable solutions of (5) by evaluating their CE, once all the saddle points of the ALI objective have been identified. However, in practice, $\mathcal{L}_{\mathrm{CE}}^\pi(\boldsymbol{\theta}, \boldsymbol{\phi})$ is intractable because we do not have access to the saddle points beforehand. Below, we propose to approximate the CE in (5) during training for both unsupervised and supervised tasks. Since $\boldsymbol{x}$ and $\boldsymbol{z}$ are symmetric in terms of CE according to (4), we use $\boldsymbol{x}$ to derive our theoretical results. Similar arguments hold for $\boldsymbol{z}$, as discussed in the Supplementary Material (SM).

## 3.2 Unsupervised Learning

In the absence of explicit probability distributions needed for computing the CE, we can bound the CE using the criterion of cycle-consistency [12]. We denote the reconstruction of $\boldsymbol{x}$ as $\hat{\boldsymbol{x}}$, via generating procedure (cycle) $\hat{\boldsymbol{x}} \sim p_{\boldsymbol{\theta}}(\hat{\boldsymbol{x}}|\boldsymbol{z}), \boldsymbol{z} \sim q_{\boldsymbol{\phi}}(\boldsymbol{z}|\boldsymbol{x}), \boldsymbol{x} \sim q(\boldsymbol{x})$. We desire that $p_{\boldsymbol{\theta}}(\hat{\boldsymbol{x}}|\boldsymbol{z})$ have high likelihood for $\hat{\boldsymbol{x}} = \boldsymbol{x}$, for the $\boldsymbol{x} \sim q(\boldsymbol{x})$ that begins the cycle $\boldsymbol{x} \rightarrow \boldsymbol{z} \rightarrow \hat{\boldsymbol{x}}$, and hence that $\hat{\boldsymbol{x}}$ be similar to the original $\boldsymbol{x}$. Lemma 3 below shows that cycle-consistency is an upper bound of the conditional entropy in (4).

**Lemma 3** *For joint distributions $p_{\boldsymbol{\theta}}(\boldsymbol{x}, \boldsymbol{z})$ or $q_{\boldsymbol{\phi}}(\boldsymbol{x}, \boldsymbol{z})$, we have*

$$H^{q_{\boldsymbol{\phi}}}(\boldsymbol{x}|\boldsymbol{z}) \triangleq -\mathbb{E}_{q_{\boldsymbol{\phi}}(\boldsymbol{x},\boldsymbol{z})}[\log q_{\boldsymbol{\phi}}(\boldsymbol{x}|\boldsymbol{z})] = -\mathbb{E}_{q_{\boldsymbol{\phi}}(\boldsymbol{x},\boldsymbol{z})}[\log p_{\boldsymbol{\theta}}(\boldsymbol{x}|\boldsymbol{z})] - \mathbb{E}_{q_{\boldsymbol{\phi}}(\boldsymbol{z})}[\mathrm{KL}(q_{\boldsymbol{\phi}}(\boldsymbol{x}|\boldsymbol{z})\|p_{\boldsymbol{\theta}}(\boldsymbol{x}|\boldsymbol{z}))]$$

$$\leq -\mathbb{E}_{q_{\boldsymbol{\phi}}(\boldsymbol{x},\boldsymbol{z})}[\log p_{\boldsymbol{\theta}}(\boldsymbol{x}|\boldsymbol{z})] \triangleq \mathcal{L}_{\mathrm{Cycle}}(\boldsymbol{\theta}, \boldsymbol{\phi}). \tag{6}$$

where $q_{\boldsymbol{\phi}}(\boldsymbol{z}) = \int d\boldsymbol{x} q_{\boldsymbol{\phi}}(\boldsymbol{x}, \boldsymbol{z})$. The proof is in the SM. Note that latent $\boldsymbol{z}$ is implicitly involved in $\mathcal{L}_{\mathrm{Cycle}}(\boldsymbol{\theta}, \boldsymbol{\phi})$ via $\mathbb{E}_{q_{\boldsymbol{\phi}}(\boldsymbol{x},\boldsymbol{z})}[\cdot]$. For the unsupervised case we want to leverage (6) to optimize the following upper bound of (5):

$$\min_{\boldsymbol{\theta}, \boldsymbol{\phi}} \max_{\boldsymbol{\omega}} \mathcal{L}_{\mathrm{ALI}}(\boldsymbol{\theta}, \boldsymbol{\phi}, \boldsymbol{\omega}) + \mathcal{L}_{\mathrm{Cycle}}(\boldsymbol{\theta}, \boldsymbol{\phi}). \tag{7}$$

Note that as ALI reaches its optimum, $p_{\boldsymbol{\theta}}(\boldsymbol{x}, \boldsymbol{z})$ and $q_{\boldsymbol{\phi}}(\boldsymbol{x}, \boldsymbol{z})$ reach saddle point $\pi(\boldsymbol{x}, \boldsymbol{z})$, then $\mathcal{L}_{\mathrm{Cycle}}(\boldsymbol{\theta}, \boldsymbol{\phi}) \rightarrow H^{q_{\boldsymbol{\phi}}}(\boldsymbol{x}|\boldsymbol{z}) \rightarrow H^{\pi}(\boldsymbol{x}|\boldsymbol{z})$ in (4) accordingly, thus (7) effectively approaches (5) (ALICE). Unlike $\mathcal{L}_{\mathrm{CE}}^{\pi}(\boldsymbol{\theta}, \boldsymbol{\phi})$ in (4), its upper bound, $\mathcal{L}_{\mathrm{Cycle}}(\boldsymbol{\theta}, \boldsymbol{\phi})$, can be easily approximated via Monte Carlo simulation. Importantly, (7) can be readily added to ALI's objective without additional changes to the original training procedure.

The cycle-consistency property has been previously leveraged in CycleGAN [12], DiscoGAN [13] and DualGAN [14]. However, in [12, 13, 14], cycle-consistency, $\mathcal{L}_{\mathrm{Cycle}}(\boldsymbol{\theta}, \boldsymbol{\phi})$, is implemented via $\ell_k$ losses, for $k = 1, 2$, and real-valued data such as images. As a consequence of an $\ell_2$-based pixel-wise loss, the generated samples tend to be blurry [8]. Recognizing this limitation, we further suggest to enforce cycle-consistency (for better reconstruction) using *fully* adversarial training (for better generation), as an alternative to $\mathcal{L}_{\mathrm{Cycle}}(\boldsymbol{\theta}, \boldsymbol{\phi})$ in (7). Specifically, to reconstruct $\boldsymbol{x}$, we specify an $\boldsymbol{\eta}$-parameterized discriminator $f_{\boldsymbol{\eta}}(\boldsymbol{x}, \hat{\boldsymbol{x}})$ to distinguish between $\boldsymbol{x}$ and its reconstruction $\hat{\boldsymbol{x}}$:

$$\min_{\boldsymbol{\theta}, \boldsymbol{\phi}} \max_{\boldsymbol{\eta}} \mathcal{L}_{\mathrm{Cycle}}^{\mathrm{A}}(\boldsymbol{\theta}, \boldsymbol{\phi}, \boldsymbol{\eta}) = \mathbb{E}_{\boldsymbol{x} \sim q(\boldsymbol{x})}[\log \sigma(f_{\boldsymbol{\eta}}(\boldsymbol{x}, \boldsymbol{x}))]$$

$$+ \mathbb{E}_{\hat{\boldsymbol{x}} \sim p_{\boldsymbol{\theta}}(\hat{\boldsymbol{x}}|\boldsymbol{z}), \boldsymbol{z} \sim q_{\boldsymbol{\phi}}(\boldsymbol{z}|\boldsymbol{x})} \log(1 - \sigma(f_{\boldsymbol{\eta}}(\boldsymbol{x}, \hat{\boldsymbol{x}}))). \tag{8}$$

Finally, the fully adversarial training algorithm for unsupervised learning using the ALICE framework is the result of replacing $\mathcal{L}_{\mathrm{Cycle}}(\boldsymbol{\theta}, \boldsymbol{\phi})$ with $\mathcal{L}_{\mathrm{Cycle}}^{\mathrm{A}}(\boldsymbol{\theta}, \boldsymbol{\phi}, \boldsymbol{\eta})$ in (7); thus, for fixed $(\boldsymbol{\theta}, \boldsymbol{\phi})$, we maximize wrt $\{\boldsymbol{\omega}, \boldsymbol{\eta}\}$.

The use of paired samples $\{\boldsymbol{x}, \hat{\boldsymbol{x}}\}$ in (8) is critical. It encourages the generators to mimic the reconstruction relationship implied in the first joint; on the contrary, the model may reduce to the basic GAN discussed in Section 3, and generate any realistic sample in $\mathcal{X}$. The objective in (8) enjoys many theoretical properties of GAN. Particularly, Proposition 1 guarantees the existence of the optimal generator and discriminator.

**Proposition 1** *The optimal generators and discriminator $\{\boldsymbol{\theta}^*, \boldsymbol{\phi}^*, \boldsymbol{\eta}^*\}$ of the objective in (8) is achieved, if and only if $\mathbb{E}_{q_{\boldsymbol{\phi}^*}(\boldsymbol{z}|\boldsymbol{x})} p_{\boldsymbol{\theta}^*}(\hat{\boldsymbol{x}}|\boldsymbol{z}) = \delta(\boldsymbol{x} - \hat{\boldsymbol{x}})$.*

The proof is provided in the SM. Together with Lemma 2 and 3, we can also show that:

**Corollary 1** *When cycle-consistency is satisfied (the optimum in (8) is achieved), (i) a deterministic mapping enforces $\mathbb{E}_{q_{\boldsymbol{\phi}}(\boldsymbol{z})}[\mathrm{KL}(q_{\boldsymbol{\phi}}(\boldsymbol{x}|\boldsymbol{z})\|p_{\boldsymbol{\theta}}(\boldsymbol{x}|\boldsymbol{z}))] = 0$, which indicates the conditionals are matched. (ii) On the contrary, the matched conditionals enforce $H^{q_{\boldsymbol{\phi}}}(\boldsymbol{x}|\boldsymbol{z}) = 0$, which indicates the corresponding mapping becomes deterministic.*

## 3.3 Semi-supervised Learning

When the objective in (7) is optimized in an unsupervised way, the identifiability issues associated with ALI are largely reduced due to the cycle-consistency-enforcing bound in Lemma 3. This means that samples in the training data have been probabilistically "paired" with high certainty, by conditionals $p_{\boldsymbol{\theta}}(\boldsymbol{x}|\boldsymbol{z})$ and $p_{\boldsymbol{\phi}}(\boldsymbol{z}|\boldsymbol{x})$, though perhaps not in the desired configuration. In real-world applications, obtaining correctly paired data samples for the entire dataset is expensive or

even impossible. However, in some situations obtaining paired data for a very small subset of the observations may be feasible. In such a case, we can leverage the small set of empirically paired samples, to further provide guidance on selecting the correct configuration. This suggests that ALICE is suitable for semi-supervised classification.

For a paired sample drawn from empirical distribution $\tilde{\pi}(\boldsymbol{x}, \boldsymbol{z})$, its desirable joint distribution is well specified. Thus, one can directly approximate the CE as

$$H^{\tilde{\pi}}(\boldsymbol{x}|\boldsymbol{z}) \approx \mathbb{E}_{\tilde{\pi}(\boldsymbol{x},\boldsymbol{z})}[\log p_{\boldsymbol{\theta}}(\boldsymbol{x}|\boldsymbol{z})] \triangleq \mathcal{L}_{\mathrm{Map}}(\boldsymbol{\theta}) \,, \qquad (9)$$

where the approximation ($\approx$) arises from the fact that $p_{\boldsymbol{\theta}}(\boldsymbol{x}|\boldsymbol{z})$ is an approximation to $\tilde{\pi}(\boldsymbol{x}|\boldsymbol{z})$. For the supervised case we leverage (9) to approximate (5) using the following minimax objective:

$$\min_{\boldsymbol{\theta},\boldsymbol{\phi}} \max_{\boldsymbol{\omega}} \; \mathcal{L}_{\mathrm{ALI}}(\boldsymbol{\theta}, \boldsymbol{\phi}, \boldsymbol{\omega}) + \mathcal{L}_{\mathrm{Map}}(\boldsymbol{\theta}). \qquad (10)$$

Note that as ALI reaches its optimum, $p_{\boldsymbol{\theta}}(\boldsymbol{x}, \boldsymbol{z})$ and $q_{\boldsymbol{\phi}}(\boldsymbol{x}, \boldsymbol{z})$ reach saddle point $\pi(\boldsymbol{x}, \boldsymbol{z})$, then $\mathcal{L}_{\mathrm{Map}}(\boldsymbol{\theta}) \to H^{\tilde{\pi}}(\boldsymbol{x}|\boldsymbol{z}) \to H^{\pi}(\boldsymbol{x}|\boldsymbol{z})$ in (4) accordingly, thus (10) approaches (5) (ALICE).

We can employ standard losses for supervised learning objectives to approximate $\mathcal{L}_{\mathrm{Map}}(\boldsymbol{\theta})$ in (10), such as cross-entropy or $\ell_k$ loss in (9). Alternatively, to also improve generation ability, we propose an adversarial learning scheme to directly match $p_{\boldsymbol{\theta}}(\boldsymbol{x}|\boldsymbol{z})$ to the paired empirical conditional $\tilde{\pi}(\boldsymbol{x}|\boldsymbol{z})$, using conditional GAN [15] as an alternative to $\mathcal{L}_{\mathrm{Map}}(\boldsymbol{\theta})$ in (10). The $\boldsymbol{\chi}$-parameterized discriminator $f_{\boldsymbol{\chi}}$ is used to distinguish the true pair $\{\boldsymbol{x}, \boldsymbol{z}\}$ from the artificially generated one $\{\hat{\boldsymbol{x}}, \boldsymbol{z}\}$ (conditioned on $\boldsymbol{z}$), using

$$\min_{\boldsymbol{\theta}} \max_{\boldsymbol{\chi}} \; \mathcal{L}_{\mathrm{Map}}^{\mathrm{A}}(\boldsymbol{\theta}, \boldsymbol{\chi}) = \mathbb{E}_{\boldsymbol{x},\boldsymbol{z} \sim \tilde{\pi}(\boldsymbol{x},\boldsymbol{z})}[\log \sigma(f_{\boldsymbol{\chi}}(\boldsymbol{x}, \boldsymbol{z})) + \mathbb{E}_{\hat{\boldsymbol{x}} \sim p_{\boldsymbol{\theta}}(\hat{\boldsymbol{x}}|\boldsymbol{z})} \log(1 - \sigma(f_{\boldsymbol{\chi}}(\hat{\boldsymbol{x}}, \boldsymbol{z})))]. \quad (11)$$

The fully adversarial training algorithm for supervised learning using the ALICE in (11) is the result of replacing $\mathcal{L}_{\mathrm{Map}}(\boldsymbol{\theta})$ with $\mathcal{L}_{\mathrm{Map}}^{\mathrm{A}}(\boldsymbol{\theta}, \boldsymbol{\chi})$ in (10), thus for fixed $(\boldsymbol{\theta}, \boldsymbol{\phi})$ we maximize wrt $\{\boldsymbol{\omega}, \boldsymbol{\chi}\}$.

**Proposition 2** *The optimum of generators and discriminator $\{\boldsymbol{\theta}^*, \boldsymbol{\chi}^*\}$ form saddle points of objective in (11), if and only if $\tilde{\pi}(\boldsymbol{x}|\boldsymbol{z}) = p_{\boldsymbol{\theta}^*}(\boldsymbol{x}|\boldsymbol{z})$ and $\tilde{\pi}(\boldsymbol{x}, \boldsymbol{z}) = p_{\boldsymbol{\theta}^*}(\boldsymbol{x}, \boldsymbol{z})$.*

The proof is provided in the SM. Proposition 2 enforces that the generator will map to the correctly paired sample in the other space. Together with the theoretical result for ALI in Lemma 2, we have

**Corollary 2** *When the optimum in (10) is achieved, $\tilde{\pi}(\boldsymbol{x}, \boldsymbol{z}) = p_{\boldsymbol{\theta}^*}(\boldsymbol{x}, \boldsymbol{z}) = q_{\boldsymbol{\phi}^*}(\boldsymbol{x}, \boldsymbol{z})$.*

Corollary 2 indicates that ALI's drawbacks associated with identifiability issues can be alleviated for the fully supervised learning scenario. Two conditional GANs can be used to boost the performance, each for one direction mapping. When tying the weights of discriminators of two conditional GANs, ALICE recovers Triangle GAN [16]. In practice, samples from the paired set $\tilde{\pi}(\boldsymbol{x}, \boldsymbol{z})$ often contain enough information to readily approximate the sufficient statistics of the entire dataset. In such case, we may use the following objective for semi-supervised learning:

$$\min_{\boldsymbol{\theta},\boldsymbol{\phi}} \max_{\boldsymbol{\omega}} \; \mathcal{L}_{\mathrm{ALI}}(\boldsymbol{\theta}, \boldsymbol{\phi}, \boldsymbol{\omega}) + \mathcal{L}_{\mathrm{Cycle}}(\boldsymbol{\theta}, \boldsymbol{\phi}) + \mathcal{L}_{\mathrm{Map}}(\boldsymbol{\theta}) \,. \qquad (12)$$

The first two terms operate on the entire set, while the last term only applies to the paired subset. Note that we can train (12) fully adversarially by replacing $\mathcal{L}_{\mathrm{Cycle}}(\boldsymbol{\theta}, \boldsymbol{\phi})$ and $\mathcal{L}_{\mathrm{Map}}(\boldsymbol{\theta})$ with $\mathcal{L}_{\mathrm{Cycle}}^{\mathrm{A}}(\boldsymbol{\theta}, \boldsymbol{\phi}, \boldsymbol{\eta})$ and $\mathcal{L}_{\mathrm{Map}}^{\mathrm{A}}(\boldsymbol{\theta}, \boldsymbol{\chi})$ in (8) and (11), respectively. In (12) each of the three terms are treated with equal weighting in the experiments if not specifically mentioned, but of course one may introduce additional hyperparameters to adjust the relative emphasis of each term.

## 4 Related Work: A Unified Perspective for Joint Distribution Matching

**Connecting ALI and CycleGAN.** We provide an information theoretical interpretation for cycle-consistency, and show that it is equivalent to controlling conditional entropies and matching conditional distributions. When cycle-consistency is satisfied, Corollary 1 shows that the conditionals are matched in CycleGAN. They also train additional discriminators to guarantee the matching of marginals for $\boldsymbol{x}$ and $\boldsymbol{z}$ using the original GAN objective in (2). This reveals the equivalence between ALI and CycleGAN, as the latter can also guarantee the matching of joint distributions $p_{\boldsymbol{\theta}}(\boldsymbol{x}, \boldsymbol{z})$ and $q_{\boldsymbol{\phi}}(\boldsymbol{x}, \boldsymbol{z})$. In practice, CycleGAN is easier to train, as it decomposes the joint distribution matching objective (as in ALI) into four subproblems. Our approach leverages a similar idea, and further improves it with adversarially learned cycle-consistency, when *high quality* samples are of interest.

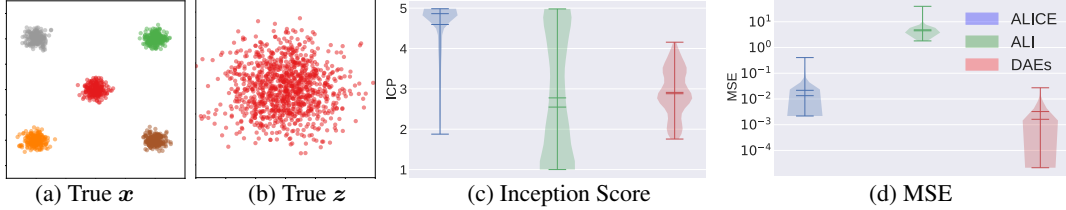

| (a) True $\boldsymbol{x}$ | (b) True $\boldsymbol{z}$ | (c) Inception Score | (d) MSE |

Figure 2: Quantitative evaluation of generation (c) and reconstruction (d) results on toy data (a,b).

**Stochastic Mapping *vs.* Deterministic Mapping**. We propose to enforce the cycle-consistency in ALI for the case when two stochastic mappings are specified as in (1). When cycle-consistency is achieved, Corollary 1 shows that the bounded conditional entropy vanishes, and thus the corresponding mapping reduces to be deterministic. In the literture, one deterministic mapping has been empirically tested in ALI's framework [4], without explicitly specifying cycle-consistency. BiGAN [10] uses two deterministic mappings. In theory, deterministic mappings guarantee cycle-consistency in ALI's framework. However, to achieve this, the model has to fit a delta distribution (deterministic mapping) to another distribution in the sense of KL divergence (see Lemma 3). Due to the asymmetry of KL, the cost function will pay extremely low cost for generating fake-looking samples [17]. This explains the underfitting reasoning in [4] behind the subpar reconstruction ability of ALI. Therefore, in ALICE, we explicitly add a cycle-consistency regularization to accelerate and stabilize training.

**Conditional GANs as Joint Distribution Matching**. Conditional GAN and its variants [15, 18, 19, 20] have been widely used in supervised tasks. Our scheme to learn conditional entropy borrows the formulation of conditional GAN [15]. To the authors' knowledge, this is the first attempt to study the conditional GAN formulation as joint distribution matching problem. Moreover, we add the potential to leverage the well-defined distribution implied by paired data, to resolve the ambiguity issues of unsupervised ALI variants [4, 10, 12, 13, 14].

## 5 Experimental Results

The code to reproduce these experiments is at `https://github.com/ChunyuanLI/ALICE`

### 5.1 Effectiveness and Stability of Cycle-Consistency

To highlight the role of the CE regularization for unsupervised learning, we perform an experiment on a toy dataset. $q(\boldsymbol{x})$ is a 2D Gaussian Mixture Model (GMM) with 5 mixture components, and $p(\boldsymbol{z})$ is chosen as a standard Gaussian, $\mathcal{N}(\mathbf{0}, \mathbf{I})$. Following [4], the covariance matrices and centroids are chosen such that the distribution exhibits severely separated modes, which makes it a relatively hard task despite its 2D nature. Following [21], to study stability, we run an exhaustive grid search over a set of architectural choices and hyper-parameters, 576 experiments for each method. We report *Mean Squared Error* (MSE) and *inception score* (denoted as ICP) [22] to quantitatively evaluate the performance of generative models. MSE is a proxy for reconstruction quality, while ICP reflects the plausibility and variety of sample generation. Lower MSE and higher ICP indicate better results. See SM for the details of the grid search and the calculation of ICP.

We train on 2048 samples, and test on 1024 samples. The ground-truth test samples for $\boldsymbol{x}$ and $\boldsymbol{z}$ are shown in Figure 2(a) and (b), respectively. We compare ALICE, ALI and Denoising Auto-Encoders (DAEs) [23], and report the distribution of ICP and MSE values, for all (576) experiments in Figure 2 (c) and (d), respectively. For reference, samples drawn from the "oracle" (ground-truth) GMM yield ICP=4.977±0.016. ALICE yields an ICP larger than 4.5 in 77% of experiments, while ALI's ICP wildly varies across different runs. These results demonstrate that ALICE is more consistent and quantitatively reliable than ALI. The DAE yields the lowest MSE, as expected, but it also results in the weakest generation ability. The comparatively low MSE of ALICE demonstrates its acceptable reconstruction ability compared to DAE, though a very significantly improvement over ALI.

Figure 3 shows the qualitative results on the test set. Since ALI's results vary largely from trial to trial, we present the one with highest ICP. In the figure, we color samples from different mixture components to highlight their correspondance between the ground truth, in Figure 2(a), and their reconstructions, in Figure 3 (first row, columns 2, 4 and 6, for ALICE, ALI and DAE, respectively). Importantly, though the reconstruction of ALI can recover the shape of manifold in $\boldsymbol{x}$ (Gaussian mixture), each individual reconstructed sample can be substantially far away from its "original" mixture component (note the highly mixed coloring), hence the poor MSE. This occurs because the adversarial training in ALI only requires that the generated samples look realistic, *i.e.,* to be located

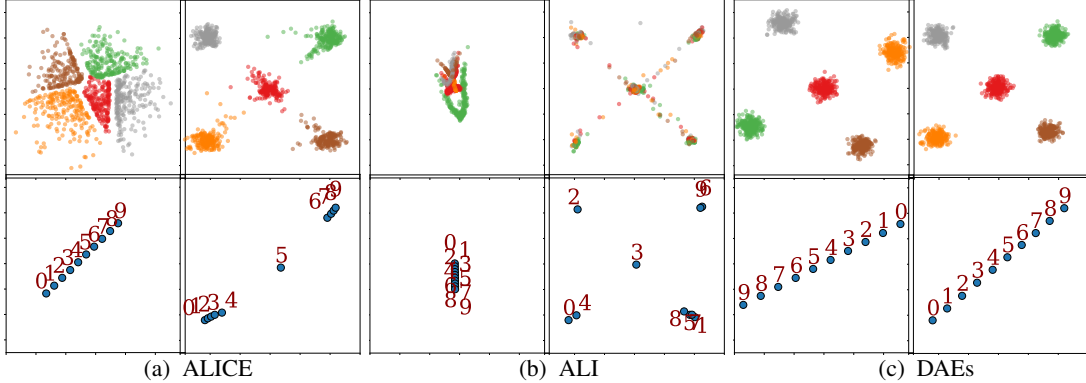

|     |     |     |
| :-: | :-: | :-: |
| (a) ALICE | (b) ALI | (c) DAEs |

Figure 3: Qualitative results on toy data. Two-column blocks represent the results of each method, with left for $z$ and right for $x$. For the first row, left is sampling of $z$, and right is reconstruction of $x$. Colors indicate mixture component membership. The second row shows reconstructions, $x$, from linearly interpolated samples in $z$.

near true samples in $\mathcal{X}$, but the mapping between observed and latent spaces ($x \to z$ and $z \to x$) is not specified. In the SM we also consider ALI with various combinations of stochastic/deterministic mappings, and conclude that models with deterministic mappings tend to have lower reconstruction ability but higher generation ability. In terms of the estimated latent space, $z$, in Figure 3 (first row, columns 1, 3 and 5, for ALICE, ALI and DAE, respectively), we see that ALICE results in a better latent representation, in the sense of *mapping consistency* (samples from different mixture components remain clustered) and *distribution consistency* (samples approximate a Gaussian distribution). The results for reconstruction of $z$ and sampling of $x$ are shown in the SM.

In Figure 3 (second row), we also investigate latent space interpolation between a pair of test set examples. We use $x_1 = [-2.2, -2.2]$ and $x_9 = [2.2, 2.2]$, map them into $z_1$ and $z_9$, linearly interpolate between $z_1$ and $z_9$ to get intermediate points $z_2, \dots, z_8$, and then map them back to the original space as $x_2, \dots, x_8$. We only show the index of the samples for better visualization. Figure 3 shows that ALICE's interpolation is smooth and consistent with the ground-truth distributions. Interpolation using ALI results in realistic samples (within mixture components), but the transition is not order-wise consistent. DAEs provides smooth transitions, but the samples in the original space look unrealistic as some of them are located in low probability density regions of the true model.

We investigate the impact of different amount of regularization on three datasets, including the toy dataset, MNIST and CIFAR-10 in SM Section D. The results show that our regularizer can improve image generation and reconstruction of ALI for a large range of weighting hyperparameter values.

## 5.2 Reconstruction and Cross-Domain Transformation on Real Datasets

Two image-to-image translation tasks are considered. (*i*) Car-to-Car [24]: each domain ($x$ and $z$) includes car images in 11 different angles, on which we seek to demonstrate the power of adversarially learned reconstruction and weak supervision. (*ii*) Edge-to-Shoe [25]: $x$ domain consists of shoe photos and $z$ domain consists of edge images, on which we report extensive quantitative comparisons. Cycle-consistency is applied on both domains. The goal is to discover the cross-domain relationship (*i.e.,* cross-domain prediction), while maintaining reconstruction ability on each domain.

**Adversarially learned reconstruction** To demonstrate the effectiveness of our fully adversarial scheme in (8) (*Joint A.*) on real datasets, we use it in place of the $\ell_2$ losses in DiscoGAN [13]. In practice, feature matching [22] is used to help the adversarial objective in (8) to reach its optimum. We also compared with a baseline scheme (*Marginal A.*) in [12], which adversarially discriminates between $x$ and its reconstruction $\hat{x}$.

The results are shown in Figure 4 (a). From top to bottom, each row shows ground-truth images, DiscoGAN (with Joint A., $\ell_2$ loss and Marginal A. schemes, respectively) and BiGAN [10]. Note that BiGAN is the best ALI variant in our grid search compasion. The proposed Joint A. scheme can retain the same crispness characteristic to adversarially-trained models, while $\ell_2$ tends to be blurry. Marginal A. provides realistic car images, but not faithful reproductions of the inputs. This explains

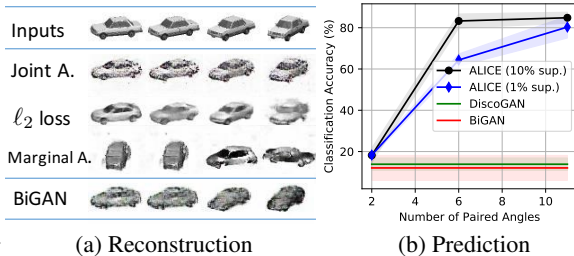

|     |     |
| :-: | :-: |
| (a) Reconstruction | (b) Prediction |

Figure 4: Results on Car-to-Car task.

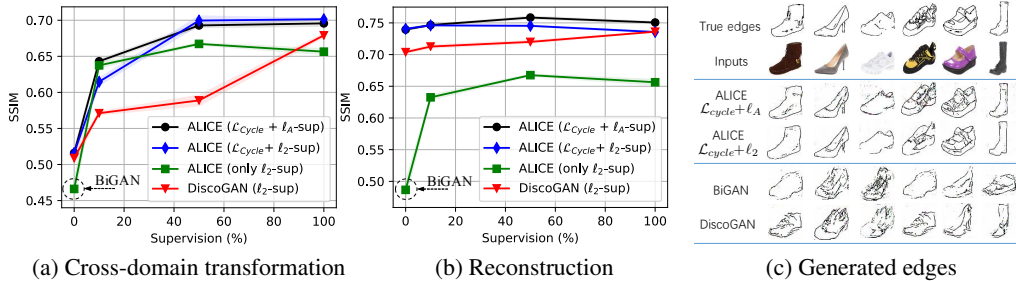

(a) Cross-domain transformation    (b) Reconstruction    (c) Generated edges

Figure 5: SSIM and generated images on Edge-to-Shoe dataset.

the observations in [12] in terms of no performance gain. The BiGAN learns the shapes of cars, but misses the textures. This is a sign of underfitting, thus indicating BiGAN is not easy to train.

**Weak supervision** The DiscoGAN and BiGAN are unsupervised methods, and exhibit very different cross-domain pairing configurations during different training epochs, which is indicative of non-identifiability issues. We leverage very weak supervision to help with convergence and guide the pairing. The results on shown in Figure 4 (b). We run each methods 5 times, the width of the colored lines reflect the standard deviation. We start with 1% true pairs for supervision, which yields significantly higher accuracy than DiscoGAN/BiGAN. We then provided 10% supervison in only 2 or 6 angles (of 11 total angles), which yields comparable angle prediction accuracy with full angle supervison in testing. This shows ALICE's ability in terms of zero-shot learning, *i.e.,* predicting unseen pairs. In the SM, we show that enforcing different weak supervision strategies affects the final pairing configurations, *i.e.,* we can leverage supervision to obtain the desirable joint distribution.

**Quantitative comparison** To quantitatively assess the generated images, we use *structural similarity* (SSIM) [26], which is an established image quality metric that correlates well with human visual perception. SSIM values are between $[0, 1]$; higher is better. The SSIM of ALICE on prediction and reconstruction is shown in Figure 5 (a)(b) for the edge-to-shoe task. As a baseline, we set DiscoGAN with $\ell_2$-based supervision ($\ell_2$-sup). BiGAN/ALI, highlighted with a circle is outperformed by ALICE in two aspects: (*i*) In the unpaired setting (0% supervision), cycle-consistency regularization ($\mathcal{L}_{\text{Cycle}}$) shows significant performance gains, particularly on reconstruction. (*ii*) When supervision is leveraged (10%), SSIM is significantly increased on prediction. The adversarial-based supervision ($\ell_A$-sup) shows higher prediction than $\ell_2$-sup. ALICE achieves very similar performance with the 50% and full supervision setup, indicating its advantage of in semi-supervised learning. Several generated edge images (with 50% supervision) are shown in Figure 5(c), $\ell_A$-sup tends to provide more details than $\ell_2$-sup. Both methods generate correct paired edges, and quality is higher than BiGAN and DiscoGAN. In the SM, we also report MSE metrics, and results on edge domain only, which are consistent with the results presented here.

**One-side cycle-consistency** When uncertainty in one domain is desirable, we consider one-side cycle-consistency. This is demonstrated on the CelebA face dataset [27]. Each face is associated with a 40-dimensional attribute vector. The results are in the Figure 8 of SM. In the first task, we consider the images $x$ are generated from a 128-dimensional Gaussian latent space $z$, and apply $\mathcal{L}_{\text{Cycle}}$ on $x$. We compare ALICE and ALI on reconstruction in Figure 8 (a)(b). ALICE shows more faithful reproduction of the input subjects. In the second task, we consider $z$ as the attribute space, from which the images $x$ are generated. The mapping from $x$ to $z$ is then attribute classification. We only apply $\mathcal{L}_{\text{Cycle}}$ on the attribute domain, and $\mathcal{L}_{\text{Map}}^{\text{A}}$ on both domains. When 10% paired samples are considered, the predicted attributes still reach 86% accuracy, which is comparable with the fully supervised case. To test the diversity on $x$, we first predict the attributes of a true face image, and then generated multiple images conditioned on the predicted attributes. Four examples are shown in Figure 8 (c).

## 6 Conclusion

We have studied the problem of non-identifiability in bidirectional adversarial networks. A unified perspective of understanding various GAN models as joint matching is provided to tackle this problem. This insight enables us to propose ALICE (with both adversarial and non-adversarial solutions) to reduce the ambiguity and control the conditionals in unsupervised and semi-supervised learning. For future work, the proposed view can provide opportunities to leverage the advantages of each model, to advance joint-distribution modeling.

**Acknowledgements**  We acknowledge Shuyang Dai, Chenyang Tao and Zihang Dai for helpful feedback/editing. This research was supported in part by ARO, DARPA, DOE, NGA, ONR and NSF.

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
