[Supplementary Material · nips_2017_supp.pdf]

# Supplementary Material of
# ALICE: Towards Understanding Adversarial Learning for Joint Distribution Matching

**Chunyuan Li[1], Hao Liu[2], Changyou Chen[3], Yunchen Pu[1], Liqun Chen[1],**
**Ricardo Henao[1] and Lawrence Carin[1]**
[1]Duke University  [2]Nanjing University  [3]University at Buffalo
http://chunyuan.li/

## A    Information Measures

Since our paper constrain correlation of two random variables using information theoretical measures, we first review the related concepts. For any probability measure $\pi$ on the random variables $\boldsymbol{x}$ and $\boldsymbol{z}$, we have the following additive and subtractive relationships for various information measures, including Mutual Information (MI), Variation of Information (VI) and the Conditional Entropy (CE).

$$\mathsf{VI}(\boldsymbol{x}, \boldsymbol{z}) = -\mathbb{E}_{\pi(\boldsymbol{z},\boldsymbol{x})}[\log \pi(\boldsymbol{x}|\boldsymbol{z})] - \mathbb{E}_{\pi(\boldsymbol{x},\boldsymbol{z})}[\log \pi(\boldsymbol{z}|\boldsymbol{x})] \tag{1}$$

$$= -\mathbb{E}_{\pi(\boldsymbol{z},\boldsymbol{x})}[\log \frac{\pi(\boldsymbol{x},\boldsymbol{z})}{\pi(\boldsymbol{x})\pi(\boldsymbol{z})} + \log \pi(\boldsymbol{x},\boldsymbol{z})] \tag{2}$$

$$= -I_\pi(\boldsymbol{x}, \boldsymbol{z}) + H_\pi(\boldsymbol{x}, \boldsymbol{z}) \tag{3}$$

$$= -\mathbb{E}_{\pi(\boldsymbol{z},\boldsymbol{x})}[\log \frac{\pi(\boldsymbol{x},\boldsymbol{z})}{\pi(\boldsymbol{x})\pi(\boldsymbol{z})} + \log \pi(\boldsymbol{x},\boldsymbol{z})] \tag{4}$$

$$= -2I_\pi(\boldsymbol{x}, \boldsymbol{z}) + H_\pi(\boldsymbol{x}) + H_\pi(\boldsymbol{z}) \tag{5}$$

### A.1    Relationship between Mutual Information, Conditional Entropy and the Negative Log Likelihood of Reconstruction

The following shows how the negative log probability (NLL) of the reconstruction is related to variation of information and mutual information. On the support of $(\boldsymbol{x}, \boldsymbol{z})$, we denote $q$ as the encoder probability measure, and $p$ as the decoder probability measure. Note that the reconstruction loss for $\boldsymbol{z}$ can be writen as its log likelihood form as $\mathcal{L}_R = -\mathbb{E}_{\boldsymbol{z}\sim p(\boldsymbol{z}),\boldsymbol{x}\sim p(\boldsymbol{x}|\boldsymbol{z})}[\log q(\boldsymbol{z}|\boldsymbol{x})]$.

**Lemma 1** *For random variables $\boldsymbol{x}$ and $\boldsymbol{z}$ with two different probability measures, $p(\boldsymbol{x}, \boldsymbol{z})$ and $q(\boldsymbol{x}, \boldsymbol{z})$, we have*

$$H_p(\boldsymbol{z}|\boldsymbol{x}) = -\mathbb{E}_{\boldsymbol{z}\sim p(\boldsymbol{z}),x\sim p(\boldsymbol{x}|\boldsymbol{z})}[\log p(\boldsymbol{z}|\boldsymbol{x})] \tag{6}$$

$$= -\mathbb{E}_{\boldsymbol{z}\sim p(\boldsymbol{z}),\boldsymbol{x}\sim p(\boldsymbol{x}|\boldsymbol{z})}[\log q(\boldsymbol{z}|\boldsymbol{x})] - \mathbb{E}_{\boldsymbol{z}\sim p(\boldsymbol{z}),x\sim p(\boldsymbol{x}|\boldsymbol{z})}\big[ \log p(\boldsymbol{z}|\boldsymbol{x}) - \log q(\boldsymbol{z}|\boldsymbol{x}) \big] \tag{7}$$

$$= -\mathbb{E}_{\boldsymbol{z}\sim p(\boldsymbol{z}),\boldsymbol{x}\sim p(\boldsymbol{x}|\boldsymbol{z})}[\log q(\boldsymbol{z}|\boldsymbol{x})] - \mathbb{E}_{p(\boldsymbol{x})}(\mathsf{KL}(p(\boldsymbol{z}|\boldsymbol{x})\|q(\boldsymbol{z}|\boldsymbol{x}))) \tag{8}$$

$$\leq -\mathbb{E}_{\boldsymbol{z}\sim p(\boldsymbol{z}),\boldsymbol{x}\sim p(\boldsymbol{x}|\boldsymbol{z})}[\log q(\boldsymbol{z}|\boldsymbol{x})] \tag{9}$$

where $H_p(\boldsymbol{z}|\boldsymbol{x})$ is the conditional entropy. From lemma 1, we have

**Corollary 1** *For random variables $\boldsymbol{x}$ and $\boldsymbol{z}$ with probability measure $p(\boldsymbol{x}, \boldsymbol{z})$, the mutual information between $\boldsymbol{x}$ and $\boldsymbol{z}$ can be written as*

$$I_p(\boldsymbol{x}, \boldsymbol{z}) = H_p(\boldsymbol{z}) - H_p(\boldsymbol{z}|\boldsymbol{x}) \geq H_p(\boldsymbol{z}) + \mathbb{E}_{\boldsymbol{z}\sim p(\boldsymbol{z}),\boldsymbol{x}\sim p(\boldsymbol{x}|\boldsymbol{z})}[\log q(\boldsymbol{z}|\boldsymbol{x})]. \tag{10}$$

Given a simple prior $p(\boldsymbol{z})$ such as isotropic Gaussian, $H(\boldsymbol{z})$ is a constant.

**Corollary 2** *For random variables $\boldsymbol{x}$ and $\boldsymbol{z}$ with probability measure $p(\boldsymbol{x}, \boldsymbol{z})$, the variation of information between $\boldsymbol{x}$ and $\boldsymbol{z}$ can be written as*

$$VI_p(\boldsymbol{x}, \boldsymbol{z}) = H_p(\boldsymbol{x}|\boldsymbol{z}) + H_p(\boldsymbol{z}|\boldsymbol{x}) \geq H_p(\boldsymbol{x}|\boldsymbol{z}) - \mathbb{E}_{\boldsymbol{z}\sim p(\boldsymbol{z}), \boldsymbol{x}\sim p(\boldsymbol{x}|\boldsymbol{z})}[\log q(\boldsymbol{z}|\boldsymbol{x})]. \quad (11)$$

# B  Proof for Adversarial Learning Schemes

The proof for cycle-consistency and conditional GAN using adversarial traning is shown below. It follows the proof of the original GAN paper: we first show the implication of optimal discriminator, and then show the corresponding optimal generator.

## B.1  Proof of Proposition 1: Adversarially Learned Cycle-Consistency for Unpair Data

In the unsupervised case, given data sample $\boldsymbol{x}$, one desirable property is reconstruction. The following game learns to reconstruct:

$$\min_{\boldsymbol{\theta}, \boldsymbol{\phi}} \max_{\boldsymbol{\omega}} \mathcal{L}(\boldsymbol{\theta}, \boldsymbol{\phi}, \boldsymbol{\omega}) = \mathbb{E}_{\boldsymbol{x}\sim q(\boldsymbol{x})}[\log \sigma(f_{\boldsymbol{\omega}}(\boldsymbol{x}, \boldsymbol{x})) + \mathbb{E}_{\boldsymbol{z}\sim q_{\boldsymbol{\phi}}(\boldsymbol{z}|\boldsymbol{x}), \hat{\boldsymbol{x}}\sim p_{\boldsymbol{\theta}}(\hat{\boldsymbol{x}}|\boldsymbol{z})} \log(1 - \sigma(f_{\boldsymbol{\omega}}(\boldsymbol{x}, \hat{\boldsymbol{x}})))]$$
$$(12)$$

**Proposition 1** *For fixed $(\boldsymbol{\theta}, \boldsymbol{\phi})$, the optimal $\boldsymbol{\omega}$ in (12) yields $f_{\boldsymbol{\omega}^*}(\boldsymbol{x}, \hat{\boldsymbol{x}}) = \mathbb{E}_{q_{\boldsymbol{\phi}}(\boldsymbol{z}|\boldsymbol{x})} p_{\boldsymbol{\theta}}(\hat{\boldsymbol{x}}|\boldsymbol{z}) = \delta(\hat{\boldsymbol{x}} - \boldsymbol{x})$.*

**Proof**  We start from a simple observation

$$\mathbb{E}_{\boldsymbol{x}\sim q(\boldsymbol{x})} \log \sigma(f_{\boldsymbol{\omega}}(\boldsymbol{x}, \boldsymbol{x})) = \mathbb{E}_{\boldsymbol{x}\sim q(\boldsymbol{x}), \hat{\boldsymbol{x}}\sim \tilde{q}(\hat{\boldsymbol{x}}|\boldsymbol{x})} \log \sigma(f_{\boldsymbol{\omega}}(\boldsymbol{x}, \hat{\boldsymbol{x}})) \quad (13)$$

when $\tilde{q}(\hat{\boldsymbol{x}}|\boldsymbol{x}) \triangleq \delta(\hat{\boldsymbol{x}} - \boldsymbol{x})$. Therefore, the objective in (12) can be expressed as

$$\mathbb{E}_{\boldsymbol{x}\sim q(\boldsymbol{x}), \hat{\boldsymbol{x}}\sim \tilde{q}(\hat{\boldsymbol{x}}|\boldsymbol{x})} \log \sigma(f_{\boldsymbol{\omega}}(\boldsymbol{x}, \hat{\boldsymbol{x}})) + \mathbb{E}_{\boldsymbol{x}\sim q(\boldsymbol{x}), \boldsymbol{z}\sim q_{\boldsymbol{\phi}}(\boldsymbol{z}|\boldsymbol{x}), \hat{\boldsymbol{x}}\sim p_{\boldsymbol{\theta}}(\hat{\boldsymbol{x}}|\boldsymbol{z})} \log(1 - \sigma(f_{\boldsymbol{\omega}}(\boldsymbol{x}, \hat{\boldsymbol{x}}))) \quad (14)$$

$$= \int_{\boldsymbol{x}} \int_{\hat{\boldsymbol{x}}} \left\{ q(\boldsymbol{x})\tilde{q}(\hat{\boldsymbol{x}}|\boldsymbol{x}) \log \sigma(f_{\boldsymbol{\omega}}(\boldsymbol{x}, \hat{\boldsymbol{x}})) + \int_{\boldsymbol{z}} q(\boldsymbol{x})q_{\boldsymbol{\phi}}(\boldsymbol{z}|\boldsymbol{x})p_{\boldsymbol{\theta}}(\hat{\boldsymbol{x}}|\boldsymbol{z}) \log(1 - \sigma(f_{\boldsymbol{\omega}}(\boldsymbol{x}, \hat{\boldsymbol{x}})))d\boldsymbol{z} \right\} d\boldsymbol{x}d\hat{\boldsymbol{x}}$$
$$(15)$$

Note that

$$\int_{\boldsymbol{z}} q(\boldsymbol{x})q_{\boldsymbol{\phi}}(\boldsymbol{z}|\boldsymbol{x})p_{\boldsymbol{\theta}}(\hat{\boldsymbol{x}}|\boldsymbol{z}) \log(1 - \sigma(f_{\boldsymbol{\omega}}(\boldsymbol{x}, \hat{\boldsymbol{x}})))d\boldsymbol{z} \quad (16)$$

$$= q(\boldsymbol{x}) \log(1 - \sigma(f_{\boldsymbol{\omega}}(\boldsymbol{x}, \hat{\boldsymbol{x}}))) \int_{\boldsymbol{z}} q_{\boldsymbol{\phi}}(\boldsymbol{z}|\boldsymbol{x})p_{\boldsymbol{\theta}}(\hat{\boldsymbol{x}}|\boldsymbol{z})d\boldsymbol{z} \quad (17)$$

$$= q(\boldsymbol{x})[\mathbb{E}_{q_{\boldsymbol{\phi}}(\boldsymbol{z}|\boldsymbol{x})} p_{\boldsymbol{\theta}}(\hat{\boldsymbol{x}}|\boldsymbol{z})] \log[1 - \sigma(f_{\boldsymbol{\omega}}(\boldsymbol{x}, \hat{\boldsymbol{x}}))] \quad (18)$$

The expression in (14) is maximal as a function of $f_{\boldsymbol{\omega}}(\boldsymbol{x}, \hat{\boldsymbol{x}})$ if and only if the integrand is maximal for every $(\boldsymbol{x}, \hat{\boldsymbol{x}})$. However, the problem $\max_t a \log(t) + b \log(1 - t)$ attains its maximum at $t = \frac{a}{a+b}$, showing that

$$\sigma(f_{\boldsymbol{\omega}^*}(\boldsymbol{x}, \hat{\boldsymbol{x}})) = \frac{q(\boldsymbol{x})\tilde{q}(\hat{\boldsymbol{x}}|\boldsymbol{x})}{q(\boldsymbol{x})\tilde{q}(\hat{\boldsymbol{x}}|\boldsymbol{x}) + q(\boldsymbol{x})\mathbb{E}_{q_{\boldsymbol{\phi}}(\boldsymbol{z}|\boldsymbol{x})} p_{\boldsymbol{\theta}}(\hat{\boldsymbol{x}}|\boldsymbol{z})} = \frac{\tilde{q}(\hat{\boldsymbol{x}}|\boldsymbol{x})}{\tilde{q}(\hat{\boldsymbol{x}}|\boldsymbol{x}) + \mathbb{E}_{q_{\boldsymbol{\phi}}(\boldsymbol{z}|\boldsymbol{x})} p_{\boldsymbol{\theta}}(\hat{\boldsymbol{x}}|\boldsymbol{z})} \quad (19)$$

For the game in (12), for which $(\boldsymbol{\theta}, \boldsymbol{\phi})$ are optimized as to most confuse the discriminator, the optimal solution for the distribution parameters $(\boldsymbol{\theta}^*, \boldsymbol{\phi}^*)$ yield $\sigma(f_{\boldsymbol{\omega}^*}(\boldsymbol{x}, \hat{\boldsymbol{x}})) = 1/2$ [1], and therefore from (19)

$$\mathbb{E}_{q_{\boldsymbol{\phi}^*}(\boldsymbol{z}|\boldsymbol{x})} p_{\boldsymbol{\theta}^*}(\hat{\boldsymbol{x}}|\boldsymbol{z}) = \delta(\boldsymbol{x} - \hat{\boldsymbol{x}}). \quad (20)$$

∎

Similarly, we can show the cycle consistency property for reconstructing $\boldsymbol{z}$ as $\mathbb{E}_{p_{\boldsymbol{\theta}^*}(\boldsymbol{x}|\boldsymbol{z})} q_{\boldsymbol{\phi}^*}(\hat{\boldsymbol{z}}|\boldsymbol{x}) = \delta(\boldsymbol{z} - \hat{\boldsymbol{z}})$.

## B.2 Proof of Proposition 2: Adversarially Learned Conditional Generation for Paired Data

In the supervised case, given the paired data sample $\pi(\boldsymbol{x}, \boldsymbol{z})$, the following game is used to conditionally generate $\boldsymbol{x}$ [2]:

$$\min_{\boldsymbol{\theta}} \max_{\boldsymbol{\omega}} \mathcal{L}(\boldsymbol{\theta}, \boldsymbol{\omega}) = \mathbb{E}_{\boldsymbol{x}, \boldsymbol{z} \sim \pi(\boldsymbol{x}, \boldsymbol{z})}[\log \sigma(f_{\boldsymbol{\omega}}(\boldsymbol{x}, \boldsymbol{z})) + \mathbb{E}_{\tilde{\boldsymbol{x}} \sim p_{\boldsymbol{\theta}}(\tilde{\boldsymbol{x}}|\boldsymbol{z})} \log(1 - \sigma(f_{\boldsymbol{\omega}}(\tilde{\boldsymbol{x}}, \boldsymbol{z})))] \quad (21)$$

To show the results, we need the following Lemma:

**Lemma 2** *The optimial generator and discriminator, with parameters $(\boldsymbol{\theta}^*, \boldsymbol{\omega}^*)$, forms the saddle points of game in* (21)*, if and only if $p_{\boldsymbol{\theta}^*}(\boldsymbol{x}|\boldsymbol{z}) = \pi(\boldsymbol{x}|\boldsymbol{z})$. Further, $p_{\boldsymbol{\theta}^*}(\boldsymbol{x}, \boldsymbol{z}) = \pi(\boldsymbol{x}, \boldsymbol{z})$*

**Proof** For the observed paired data $\pi(\boldsymbol{x}, \boldsymbol{z})$, we have $p(\boldsymbol{z}) = \pi(\boldsymbol{z})$, where $\pi(\boldsymbol{z})$ is marginal empirical distribution of $\boldsymbol{z}$ for the paired data.

Also, $\pi(\tilde{\boldsymbol{x}}|\boldsymbol{z}) = \delta(\tilde{\boldsymbol{x}} - \boldsymbol{x})$ when $\tilde{\boldsymbol{x}}$ is paired with $\boldsymbol{z}$ in the dataset. We start from the observation

$$\mathbb{E}_{\boldsymbol{x}, \boldsymbol{z} \sim \pi(\boldsymbol{x}, \boldsymbol{z})} \log \sigma(f_{\boldsymbol{\omega}}(\boldsymbol{x}, \boldsymbol{z})) = \mathbb{E}_{\boldsymbol{z} \sim p(\boldsymbol{z}), \tilde{\boldsymbol{x}} \sim \pi(\tilde{\boldsymbol{x}}|\boldsymbol{z})} \log \sigma(f_{\boldsymbol{\omega}}(\tilde{\boldsymbol{x}}, \boldsymbol{z})) \quad (22)$$

Therefore, the objective in (21) can be expressed as

$$\mathbb{E}_{\boldsymbol{x} \sim p(\boldsymbol{z}), \tilde{\boldsymbol{x}} \sim \pi(\tilde{\boldsymbol{x}}|\boldsymbol{z})} \log \sigma(f_{\boldsymbol{\omega}}(\tilde{\boldsymbol{x}}, \boldsymbol{z})) + \mathbb{E}_{\boldsymbol{z} \sim p(\boldsymbol{z}), \tilde{\boldsymbol{x}} \sim p_{\boldsymbol{\theta}}(\tilde{\boldsymbol{x}}|\boldsymbol{z})} \log(1 - \sigma(f_{\boldsymbol{\omega}}(\tilde{\boldsymbol{x}}, \boldsymbol{z}))) \quad (23)$$

This integral is maximal as a function of $f_{\boldsymbol{\omega}}(\boldsymbol{x}, \boldsymbol{z})$ if and only if the integrand is maximal for every $(\boldsymbol{x}, \boldsymbol{z})$. However, the problem $\max_t a \log(t) + b \log(1 - t)$ attains its maximum at $t = \frac{a}{a+b}$, showing that

$$\sigma(f_{\boldsymbol{\omega}^*}(\boldsymbol{x}, \boldsymbol{z})) = \frac{p(\boldsymbol{x})\pi(\boldsymbol{x}|\boldsymbol{z})}{p(\boldsymbol{x})\pi(\boldsymbol{x}|\boldsymbol{z}) + p(\boldsymbol{z})p_{\boldsymbol{\theta}}(\boldsymbol{x}|\boldsymbol{z})} = \frac{\pi(\boldsymbol{x}|\boldsymbol{z})}{\pi(\boldsymbol{x}|\boldsymbol{z}) + p_{\boldsymbol{\theta}}(\boldsymbol{x}|\boldsymbol{z})} \quad (24)$$

or equivalently, the optimum generator is $p_{\boldsymbol{\theta}^*}(\boldsymbol{x}|\boldsymbol{z}) = \pi(\boldsymbol{x}|\boldsymbol{z})$. Since $q(\boldsymbol{x}) = \pi(\boldsymbol{x})$, we further have $p_{\boldsymbol{\theta}^*}(\boldsymbol{x}, \boldsymbol{z}) = \pi(\boldsymbol{x}, \boldsymbol{z})$. Similarly, for conditional GAN of $\boldsymbol{z}$, we can show that is $q_{\boldsymbol{\phi}^*}(\boldsymbol{z}|\boldsymbol{x}) = \pi(\boldsymbol{z}|\boldsymbol{x})$ and $q_{\boldsymbol{\phi}^*}(\boldsymbol{x}, \boldsymbol{z}) = \pi(\boldsymbol{x}, \boldsymbol{z})$ for the Combining them, we show that $p_{\boldsymbol{\theta}^*}(\boldsymbol{x}, \boldsymbol{z}) = \pi(\boldsymbol{x}, \boldsymbol{z}) = q_{\boldsymbol{\phi}^*}(\boldsymbol{x}, \boldsymbol{z})$. ∎

# C More Results on the Toy Data

## C.1 The detailed setup

The 5-component Gaussian mixture model (GMM) in $\boldsymbol{x}$ is set with the means $(0, 0), (2, 2), (-2, 2), (2, -2), (-2, -2)$, and standard derivation $0.2$. The Isotropic Gaussian in $\boldsymbol{z}$ is set with mean $(0, 0)$ and standard derivation $1.0$.

We consider various network architectures to compare the stability of the methods. The hyperparameters includes: the number of layers and the number of neurons of the discriminator and two generators, and the update frenquency for discriminator and generator. The grid search specification is summarized in Table 1. Hence, the total number of experiments is $2^3 \times 2^3 \times 3^2 = 576$.

A generalized version of the *inception score* is calculated, $\text{ICP} = \mathbb{E}_{\boldsymbol{x}} \text{KL}(p(y)||p(y|\boldsymbol{x}))$, where $\boldsymbol{x}$ denotes a generated sample and $y$ is the label predicted by a classifier that is trained off-line using the entire training set. It is also worth noting that although we inherit the name "inception score" from [3], our evaluation is not related to the "inception" model trained on ImageNet dataset. Our classifier is a regular 3-layer neural nets trained on the dataset of interest, which yields $100\%$ classification accuracy on this toy dataset.

## C.2 Reconstruction of $z$ and sampling for $x$

We show the additional results for the econstruction of $\boldsymbol{z}$ and sampling for $\boldsymbol{x}$ in Figure 1. ALICE shows good sampling ability, as it reflects the Guassian characteristics for each of 5 components, while ALI's samples tends to be concentrated, reflected by the shrinked Guassian components. DAE learns an indentity mapping, and thus show weak generation ability.

## C.3 Summary of the four variants of ALICE

ALICE is a general CE-based framework to regularize the objectives of bidiretional adversarial training, in order to obtain desirable solutions. To clearly show the versatility of ALICE, we summarize its four variants, and test their effectivenss on toy datasets.

In unsupervised learning, two forms of cycle-consistency/reconstruction are considered to bound CE:

|     (a) ALICE     |     (b) ALI     |     (c) DAEs     |

Figure 1: Qualitative results on toy data. Every two columns indicate the results of a method, with left space as reconstruction of $z$ and right space as sampling in $x$, respectively.

- **Explicit cycle-consistency**: Explicitly specified $\ell_k$-norm for reconstruction;
- **Implicit cycle-consistency**: Implicitly learned reconstruction via adversarial training

In semi-supervised learning, the pairwise information is leveraged in two forms to approximate CE:

- **Explicit mapping**: Explicitly specified $\ell_k$-norm mapping (*e.g.,* standard supervised losses);
- **Implicit mapping**: Implicitly learned mapping via adversarial training

**Disucssion** (*i*) Explicit methods such as $\ell_k$ losses ($k = 1, 2$): The similarity/quality of the reconstruction to the original sample is measured in terms of $\ell_k$ metric. This is easy to implement and optimize. However, it may lead to visually low quality reconstruction in high dimensions. (*ii*) Implicit methods via adversarial training: it essentially requires the reconstruction to be close to the original sample in terms of $\ell_0$ metric (see Section 3.3 of [4]: Adversarial feature learning). It theoretically guarantees perfect reconstruction, however, this is hard to achieve in practice, espcially in high dimension spaces.

**Results** The effectivenss of these algorithms are demonstrated on toy data of low dimension in Figure 2. The unsupervised variants are tested in the same toy dataset described above, the results are in Figure 2 (a)(b). For the supervised variants, we create a toy dataset, where $z$-domain is 2-component GMM, and $x$-domain is 5-component GMM. Since each domain is symmtric, ambiguity exists when Cycle-GAN variants attempt to discover the relationship of the two domains in pure unsupervised setting. Indeed, we observed random switching of the discoverd corresponded components in different runs of Cycle-GAN. By adding a tiny fraction of pairwise information (a cheap way to specify the desirable relationship ), we can easily learn the correct correspondences for the entire datasets. In Figure 2 (c)(d), 5 pairs (out of 2048) are pre-specified: the points $[0, 0], [1, 1], [-1, -1], [1, -1], [-1, 1]$ in $x$-domain are paired with the points in $z$-domain with opposite signs. Both explicit and implicit ALICE find the correct pairing configurations for other unlabeled samples. This inspires us to manually labeling the relations for a few samples between domains, and use ALICE to automatically control the full datasets pairing for the real datasets. One example is shown on Car2Car dataset.

### C.4 Comparisons of ALI with stochastic/deterministic mappings

We investigate the ALI model with different mappings:

- **ALI:** two stochastic mappings;
- **ALI$^-$:** one stochastic mapping and one deterministic mapping;
- **BiGAN:** two deterministic mappings.

We plot the histogram of ICP and MSE in Fig. 3, and report the mean and standard derivation in Table 2. In Fig. 4, we compare their reconstruction and generation ability. Models with deterministic mapping have higher recontruction ability, while show lower sampling ability.

**Comparison on Reconstruction** Please see row 1 and 2 in Fig. 4. For reconstruction, we start from one sample (red dot), and pass it through the cycle formed by the two mappings 100 times. The resulted reconstructions are shown as blue dots. The reconstructed samples tends to be concentrated with more deterministic mappings.

**Comparison on Sampling** Please see row 3 and 4 in Fig. 4. For sampling, we first draw 1024 samples in each domain, and pass them through the mappings. The generated samples are colored as the index of Gaussian component it comes from in the original domain.

(a) Explicit Cycle-Consistency (L₂ loss)

(b) Implicit Cycle-Consistency (Adversarial loss)

(c) Explicit Mapping (L₂ loss)

(d) Implicit Mapping (Adversarial loss)

Figure 2: Results of four variants of ALICE on toy datasets.

Table 1: Grid search specification.

| Settings | Values |
|---|---|
| Number of layers | $[2, 3]$ |
| Number of neurons | $[256, 512]$ |
| Update frenquency | $[1, 3, 5]$ |

Table 2: Testing MSE and ICP on toy dataset..

| Method | MSE | ICP |
|---|---|---|
| ALICE | $0.022 \pm 0.029$ | $\mathbf{4.595 \pm 0.604}$ |
| ALI | $4.856 \pm 2.920$ | $2.776 \pm 1.516$ |
| ALI$^-$ | $3.888 \pm 7.343$ | $3.420 \pm 1.299$ |
| BiGAN | $2.399 \pm 3.605$ | $3.712 \pm 1.278$ |
| DAEs | $\mathbf{0.003 \pm 0.004}$ | $2.913 \pm 0.004$ |

(a) Inception Score

(b) MSE

Figure 3: Quantitative results on toy data.

Figure 4: Comparison with bidirectional GAN models with different stochastic or deterministic mappings. The 1st row is the reconstruction of $z$, and the 2nd row is the reconstruction of $x$. In these two rows, the red dot is the original data point, the blue dots are the reconstruction. The 3rd row is the sampling of $z$, and 4th row is the sampling of $x$. and 5th row is the reconstruction for $x$. In the 3rd row, colors of the generated $z$ indicate the component of $x$ that $z$ conditions on.

# D   More Results on the Effectiveness of CE Regularizers

We investigate the effectiveness and impact of the proposed cycle-consistency regularizer (explicit $\ell_2$ norm) on 3 datasets, including the toy dataset, MNIST and CIFAR-10. A large range of weighting hyperparameter $\lambda$ is tested. The inception scores on toy and MNIST datasets are evaluted by the pre-trained "perfect" classifiers of these datasets, respectively, while inception scores on CIFAR is based on ImageNet. The results for different $\lambda$ are shown in Figure 5, and the best performance is sumarized in Table 3.

Table 3: Compariso on real datasets.

|  | Image generation (ICP ↑) | | Image reconstruction (MSE ↓) | |
|---|---|---|---|---|
| Settings | ALI | ALICE ($\lambda = 1$) | ALI | ALICE ($\lambda = 10^{-6}$) |
| MNIST | $8.749 \pm 0.09$ | $\mathbf{9.279} \pm 0.07$ | $0.4803 \pm 0.100$ | $\mathbf{0.0803} \pm 0.007$ |
| CIFAR | $5.93 \pm 0.0437$ | $\mathbf{6.015} \pm 0.0284$ | $0.672 \pm 0.1129$ | $\mathbf{0.4155} \pm 0.2015$ |

(a) Toy dataset: image generation

(b) Toy dataset: image reconstruction

(c) MNIST: image generation

(d) MNIST: image reconstruction

(e) CIFAR: image generation

(f) CIFAR: image reconstruction

Figure 5: Impact of the proposed cycle-consistency regularizer. The "perfect" performance is shown as a solid line, the ALI (*i.e.*, without CE regularizer) performance is a dash line. ALICE with different levels of regularization are shown as light blue dots, and best performance of ALICE is shown as the dot with a dark blue circle.

# E  More Details on Real Data Experiments

## E.1  Car to Car Experiment

**Setup** The dataset [5] consists of rendered images of 3D car models with varying azimuth angles at $15°$ intervals. 11 views of each car are used. The dataset is split into train set ( $169 \times 11 = 1859$ images) and test set ( $14 \times 11 = 154$ images), and further split the train set into two groups, each of which is used as A domain and B domain samples. To evaluate, we trained a regressor and a classifier that predict the azimuth angle using the train set. We map the car image from one domain to the other, and then reconstruct to the original domain. The cycle-consistency is eveluted as the prediction accuracy of the reconstructed images.

Table 4 shows the MSE and prediction accuracy by leverage the supervision in different number of angles. To further demonstrate that we can easily control the correspondence configuration by designing the proper supervision, we use ALICE to enforce coherent supervsion and opposite supervision, respectively. Only $1\%$ supervison information is used in each angle. We translated images in the test set using each of the three trained models, and azimuth angles were predicted using the regressor for both input and translated images. In Table 5, we show the cross domain relationship discovered by each method. X and Y axis indicates predicted angles of original and transformed cars, respectively. All three plots are results at the 10th epoch. Scatter points with supervision are more concentrated on the diagnals in the plots, which indicates higher prediction/correlation. The learning curves are shown in Table 5(d). The Y axis indicate the RMSE in angle prediction. We see that very weak supervision can largely imporve the convergence results and speed. Example and comparison arre shown in Figure6.

Table 4: ACC and MSE in prediction on car translation. The top four methods are our methods reported in the format of #Angle (supervison%).

| Methods | MSE | ACC (%) |
|---------|-----|---------|
| 11 (1%) | 438.71±5.43 | 80.32±5.30 |
| 11 (10%) | **366.74**±0.38 | **84.83**±2.68 |
| 6 (10%) | 380.61±4.94 | 83.27±3.37 |
| 2 (10%) | 656.28±20.9 | 16.20±3.50 |
| DiscoGAN | 712.20±14.6 | 13.86±3.00 |
| BiGAN | 790.13±15.0 | 12.07±4.03 |

Table 5: The scatter plots on car2car.

(a) DiscoGAN

(b) ALICE: coherent sup.

(c) ALICE: opposite sup.

(d) Learning curves

Figure 6: Cross-domain relationship discovery with weakly supervised information using ALICE.

## E.2 Edge-to-Shoe Dataset

The MSE results on cross-domain prediction and one-domain reconstruction are shown in Figure 7.

(a) Cross-domain transformation on both sides

(b) Reconstruction on both sides

(c) Cross-domain transformation on both sides

(d) Reconstruction on on both sides

(e) Cross-domain transformation on edges

(f) Reconstruction on edges

(g) Cross-domain transformation on edges

(h) Reconstruction on edges

Figure 7: SSIM and MSE on Edge-to-Shoe dataset. Top 2 rows are results reported for both domains, and the bottom 2 rows are results for edge domain only.

### E.3 Celeba Face Dataset

Reconstruction results on the validation dataset of Celeba dataset are shown in Figure 8. ALI results are from the paper [6]. ALICE provides more faithful reconstruction to the input subjects. As a trade-off between theoretical optimum and practical convergence, we employ feature matching, and thus our results exhibits slight bluriness characteristic.

(a) ALICE          (b) ALI

(c) Generated faces.

Figure 8: Reconstruction of (a) ALICE and (b) ALI. Odd columns are original samples from the validation set and even columns are corresponding reconstructions. (c) Generated faces (even rows), based on the predicted attributes of the real face image (odd row).

### E.4 Real applications: Edges to Cartoon

We demonstrate the potential real applications of ALICE algorithms on the task of sketch to cartoon. We built a dataset by collecting frames from Disney's film `Alice in Wonderland`. A large image size $256 \times 256$ is considered. The training dataset consists of two domains: cartoon images and edges images, where the edges are created via holistically-nested edge detection [7] on their true cartoon images. The image content is about either of two characters in the film: Alice or White Rabbit. Therefore, each domain exhibits two modes. 52 images are collected in each domain. The one-to-one image correspondence between two domain is unknown, the goal is to efficiently generate realistic cartoon images for animation, based on the edges.

CycleGAN is an unsupervised learning algorithm. Since we have shown its equivalence to ALI/BiGAN (see Related Work), and its superiority in terms of stability. We derive our weakly supervised ALICE algorithm on this dataset as: ($i$) CycleGAN for the unpaired data, and ($ii$) explicit $\ell_0$ loss and/or implicit conditional GAN loss for the paired samples. Note that only one pair are randomly chosen for each character. We summarized the results:

- **ALICE converges faster than CycleGAN:** . The generated images after 6K iterations are show in Figure 9. CycleGAN generates images with mixed colors (for example, the clothes of Rabbit), while ALICE clearly paint colors in different regions.

- **ALICE enbles desirable generated images (with better generalization):** The generated images after 10K iterations are show in Figure 10. We generated image based on slightly different edges: more background and details on the character. CycleGAN gets confused when identifying the character, thus inconsistently paint the wrong color to Rabbit, while explicit ALICE still generates correct color.

(a) Real Cartoon

(b) Real Edges

(c) Generated cartoon images via explicit ALICE

(d) Generated cartoon images via implicit ALICE

(e) Generated cartoon images via CycleGAN

Figure 9: Generaed cartoon images (conditioned on training edges in (b)) after 6K iterations of different algorithms: (c) explicit ALICE, (d) implicit ALICE and (e) CycleGAN.

(a) Real Cartoon

(b) Real Edges

(c) Generated cartoon images via explicit ALICE

(d) Generated cartoon images via implicit ALICE

(e) Generated cartoon images via CycleGAN

Figure 10: Generaed cartoon images (conditioned on more detailed edges in (b)) after 10K iterations of different algorithms: (c) explicit ALICE, (d) implicit ALICE and (e) CycleGAN.