[Reviews · NeurIPS 2017]

Reviewer 1



Rather than providing an understanding of joint-distribution matching using adversarial network, the paper essentially proposes two specific algorithms based on cycle-consistency (one unsupervised, one semi-supervised) using an approach that casts interesting lights on the existing joint-distribution matching algorithms and in particular conditional entropy regularization. Besides a section discussing these similarities in detail, the authors also provide extensive experiments. I found this paper generally insightful and well thought. People unfamiliar with the topic may find it hard to read, especially because equation (5) is not really meaningful (since \pi is a product of the optimization process, it cannot be part of its specification.) However I can see what is meant by the authors, and it makes sufficient sense in my mind. I am not familiar enough with the most recent result to asses the quality of the empirical results. The experimental process however seems serious.

Reviewer 2



Adversarially Learned Inference (a.k.a. Adversarial Feature Learning) is an interesting extension to GANs, which can be used to train a generative model by learning generator G(z) and inference E(x) functions, where G(z) maps samples from a latent space to data and E(x) is an inference model mapping observed data to the latent space. This model is trained adversarially by jointly training E(x) and G(z) with a discriminator D(x,z) which is trained to distinguish between real (E(x), x) samples and fake (z, G(z)) samples. This is an interesting approach and has been shown to generate latent representations which are useful for semi-supervised learning. The authors highlight an issue with the ALI model, by constructing a small example for which there exist optimal solutions to the ALI loss function which have poor reconstruction, i.e. G(E(x)) can be very different to x. This identifiability issue is possible because the ALI objective imposes no restrictions on the conditional distributions of the generator and inference models. In particular, this allows for solutions which violate a condition known as cycle consistency, which is considered a desirable property in unsupervised learning. They propose adding a cross-entropy term to the loss, which enforces a constraint on one of the two conditionals. The ALICE training objective is shown to yield better reconstructions than ALI on toy data. They also evaluate on the inception score, a metric proposed in Salimans et al. (2017), which also shows better performance for the model trained with the ALICE objective. The description of how the model can be applied to semi-supervised learning seemed a little strange to me. The authors consider a case where the latent space z corresponds to the labels and semi-supervised learning involves training a model with the ALICE objective, when some paired (z,x) samples are available. They claim that these labelled samples can be leveraged to address the identifiability issues in ALI. While this setting holds for the experiments they present in section 5.2, a more common application of this kind of model to semi-supervised learning is to use the inference network to generate features which are then used for classification. The results in the ALI paper follow this approach and evaluate ALI for semi-supervised learning on some standard benchmarks (CIFAR10 and SVHN). This is possibly the strongest evidence in the ALI paper that the model is learning a useful mapping from data space to latent space, so it would be useful to see the same evaluation for ALICE. In the ALI and BIGAN papers, it is shown that cycle consistency is satisfied at the optimum when the E and G mappings are deterministic. Similarly, in this paper the authors show that the optimal solution to a model trained with cross-entropy regularisation has deterministic mappings. In lines 228-233, the authors suggest that a model trained with deterministic mappings is likely to underfit and taking the alternative approach of training a model with stochastic mappings and a cross-entropy regularisation gives faster and more stable training. While this is an interesting claim, it was not immediately clear to me why this would be the case from the description provided in the paper. This paper is on an interesting topic, highlighting a significant issue with the ALI training objective and proposing a solution using cross-entropy regularisation. It would be more convincing if the authors had evaluated their model on the same set of experiments as the original ALI paper, especially the semi-supervised learning benchmarks, and provided more justification for the use of cross-entropy regularisation over deterministic mappings

Reviewer 3



GANs have been receiving a lot of attention lately. They are good at generating samples using an adversarial approach. However they cannot infer the mapping from data to the latent space. One solution to this has been to use adversarial learning to distinguish between pairs from two joint distributions: the model joint and the joint defined as the product of the approximate posterior and the empirical data distribution. This approach has been used in several works one of which is ALI (Dumoulin et al. 2016). This paper casts many existing works as performing joint-distribution matching. It also discusses and proposes a solution to the identifiability issue that arises in these settings using entropy regularization. Few remarks regarding this paper are below. 1. Although I found that the experiments were not fully developed and seemed "toy", the issue raised by this paper, namely the identifiability problem in joint distribution matching is interesting. Unifying several existing works as performing this same joint distribution matching is interesting. The entropy regularization approach is interesting. I only wish the experiments were carried out more thoroughly. 2. It was not clear to me what type of restrictions are being referred to in the paper (example at the bottom of page 2). Discussing the nature of these restrictions would have been a good insight. 3. The paper does not solve the identifiability issue that arises because one latent sample z may lead to any element in the sample space. What it does is use an auxiliary discriminator that distinguishes reconstructed data from real data. This procedure, according to the paper is equivalent to entropy regularization. However this still does not ensure that one latent sample z leads to one data sample x from the data space. This would happen only in the optimal case which is usually never attained in adversarial approaches that are minimax problems. This brings back the question of what types of restrictions should be enforced to ensure identifiability. 4. Some notations are not defined in the paper. See for example section 3.2. There are also some typos: for example page 5 "p_{\theta}(x | z) is an approaximation to \pi(x , z)" should be \pi(x | z).